# The Effect of High Voltage Electrical Discharge on the Physicochemical Properties and the Microbiological Safety of Rose Hip Nectars

**DOI:** 10.3390/foods11050651

**Published:** 2022-02-23

**Authors:** Nela Nedić Tiban, Mirela Šimović, Martina Polović, Antonija Šarić, Ivana Tomac, Petra Matić, Lidija Jakobek

**Affiliations:** 1Faculty of Food Technology Osijek, Josip Juraj Strossmayer University of Osijek, Franje Kuhača 18, 31000 Osijek, Croatia; mrukavina1@ptfos.hr (M.P.); antonija.saric@ptfos.hr (A.Š.); ivana.tomac@ptfos.hr (I.T.); petra.matic@ptfos.hr (P.M.); lidija.jakobek@ptfos.hr (L.J.); 2Department of Health Ecology, Teaching Institute of Public Health Osijek-Baranja County, Drinska 8, 31000 Osijek, Croatia; mirela.simovic1@gmail.com

**Keywords:** rose hip, nectar, ascorbic acid, phenol content, colour, microbiological safety

## Abstract

Although neglected as an industrial raw material, rose hip has been important for both nutritional and medical purposes for centuries. The main goal of this study was to propose a rapid and inexpensive non-thermal technique such as high voltage electrical discharge (HVED) to preserve valuable rose hip bioactive compounds, towards the development of high-quality products, including low-calorie products. The objective of this work was to evaluate the effects of HVED on the physicochemical properties and the microbiological safety of rose hip nectar formulations and, for comparison, on a pasteurised sample. Physicochemical analysis proved that rose hip pulp and the prepared nectars were valuable sources of polyphenols and ascorbic acid with high antioxidant activity. The HVED technique had minimal effects on the quality characteristics of the nectars under the different process conditions (50, 100 Hz; 10, 15, 20 min). In addition, the pasteurised nectar showed the greatest loss of ascorbic acid (54%) and phenolic compounds (40%). The microbiological quality of nectars was examined immediately after preparation/treatment and after 6 and 12 days of storage at 4 °C. In addition to the pasteurised sample, HVED-treated rose hip nectar prepared from microwave-blanched puree with extended shelf life had satisfactory microbiological safety after storage.

## 1. Introduction

Native species growing in their natural ecosystems could be used as new food, valuable natural materials and products. In recent years, medical interest in the subgenus *Rosa*, section *Caninae*, has increased considerably due to recent research investigating its potential use for inflammatory disorders, diabetes, obesity, hepatotoxicity and many other diseases [1]. The morphological, pomological and genetic variability of the genotypes of *Rosa canina* Laxa, wild or dog rose, is spread throughout Croatia in the form of spontaneous populations and ecotypes [2]. The fruit of the dog rose ‘rose hip’ is characterised by a series of bioactive compounds with high antioxidant and antimicrobial activities, which are used in food and in the cosmetic and pharmaceutical industries, among others. Even though rose hip has been collected since ancient times, today there is almost no cultivation in the Republic of Croatia. Recently, small orchards of this species have been planted, but there are many possibilities for the production of this undemanding and high-quality fruit tree [3]. Among the valuable phytochemicals contained in rose hip, phenolic compounds and ascorbic acid stand out. It is known that rose hip has the highest vitamin C content of all fruits and vegetables. Rose hip contains sugars, organic acids, pectins, other vitamins and minerals, pigments, tocopherols, tannins, amino acids and essential oils [4]. It is generally not consumed in fresh form, but it is used to make jams, teas, juices, syrups, wines, etc. [5].

Due to their health benefits and a growing consumer awareness of health issues, fruit juices have become an essential part of the human diet. Global demand for high quality fruit juices is increasing, which has encouraged the development of new technologies that do not compromise the quality of the fruit. The effect of heat leads to the loss of compounds responsible for sensory and nutritional properties, during thermal processing and product storage [6]. Blanching is a short-term heat pretreatment used to inactivate enzymes, preserve colour and avoid associated changes during further processing into juices [7]. Dry blanching, such as microwave blanching, is one of the emerging technologies that appears to offer better nutrient retention over time and zero wastewater production. It is an alternative to traditional blanching with water, with the goal of minimising the loss of water-soluble compounds and maintaining the value of the processed fruit [8].

Current alternative non-thermal commercial techniques are expensive due to investment, increased labour and product type limitation. The high voltage electrical discharge (HVED) treatment is particularly attractive because it does not require extreme process conditions compared to high pressure treatment and other preservation methods [9]. While the HVED treatment is widely used in industrial processes, its potential has not yet been exploited in the food sector [10]. The simplicity of the HVED design and its low power requirements reduce capital and operating costs and it can be used to treat both solid and liquid foods. Plasma chemistry is a complex science used to understand the interaction between food components involving numerous species in a myriad of chemical reactions occurring in different time scales [11]. Free electrons produced by a strong electric field interact with nearby gas molecules to form a quasi-stable charged gas species or plasma. The hydroxyl radicals NO and NO_2_, and other resulting reactive gas species, may have bactericidal, fungicidal and sporicidal characteristics [12,13]. Inactivation of both pathogenic and spoilage microorganisms could enable minimally processed HVED foods with extended shelf life. The application of cold plasma can ensure better qualitative properties and a higher nutritional value in juices [14].

There are conflicting studies associated with the loss of bioactive components in juices after plasma treatment. As for vitamins in processed juices, the effect of plasma on vitamin C content has been the most researched [15]. Generally, the amount of vitamin C in juice is used as an indicator of quality and as a criterion for other nutritional components [6,11]. The amounts of phenols in the products may change during cold plasma treatment. Plasma treatment of orange juice and white grape juice showed a decrease in total phenols [16,17]. Studies that showed a significant increase in phenol content in cashew apple, tomato and sour cherry juices were also reported [18,19,20]. These differences in studies highlight the need for research to better understand the effects of plasma on polyphenols. Another important quality parameter that should be measured after the HVED treatment is the antioxidant capacity of juices. However, the effects of the plasma on polyphenols and the antioxidant activity of juices are still unclear, and they need further investigation.

Some studies show the antibacterial and antifungal effect of cold plasma on pasteurised juices previously contaminated with pathogenic microorganisms [18,21,22,23,24,25]. Shi et al. [26] studied the application of plasma to freshly squeezed orange juice and confirmed that plasma can extend the shelf life of the juice. Plasma treatment has been shown to be effective against pathogens such as *Staphylococcus aureus*, *Escherichia coli*, *Salmonella* Typhimurium and *Listeria monocytogenes* [11]. Xu et al. [13] concluded that cold plasma treatment could be an effective non-thermal method for controlling or potentially eliminating *Salmonella* in orange juice. Dasan and Boyaci [24] observed a reduction in the number of bacteria by applying plasma to apple, orange and tomato juices. Apple juice treatments with cold plasma and the inactivation of *Zygosaccharomyces rouxii* have been studied by Xiang et al. [21] and *Citrobacter freundii* by Surowsky et al. [22].

Most published research has focused on microbial decontamination, whereas there are few studies on the effects of HVED processing on the microbiological safety and quality characteristics of juices as minimally processed products [19,26]. However, there are no data on the effects of HVED on the quality and safety of rose hip nectar, which is the focus of this study.

Therefore, the aim of this study is to investigate the effect of high voltage electrical discharge on the physicochemical properties and microbiological quality and safety of rose hip nectar formulations. Subsequently, the compliance of the samples with the applicable microbiological criteria is determined with respect to aerobic mesophilic bacteria, *Enterobacteriaceae*, *Salmonella* spp., *Escherichia coli, Listeria monocytogenes* and yeasts and moulds.

## 2. Materials and Methods

### 2.1. Materials

The rose hip fruit (*Rosa canina* Laxa) was collected by hand in the region of Nova Gradiška (Slavonia, Croatia) in September 2020. The rose hip (uniform in shape and colour) was cleaned, washed, towel-dried and froze immediately after harvesting. The samples were stored at −18 °C.

All chemicals used were of analytical grade. The following ingredients were also used: commercial sugar (sucrose), low-calorie sweetener: isomalt 99.2% and steviol glycoside 0.8% (Sweet Stevia, Vitalia, Skopje, N. Macedonia) and pectin (Combi AC 10, Herbstreith & Fox, Neuenbürg, Germany).

### 2.2. Characterisation of Rose Hip Pulp

After separating the waste (parts of the stem; and the remains of the cup and the skin), the fruit was cut lengthwise, removing tiny hairs and seeds. Then it was crushed with a stick blender (Braun, Frankfurt, Germany).

The following analyses of rose hip pulp were performed. A total dry matter of pulp (TDM, %) was obtained by vacuum drying (VS-50 SC, Kambič, Semič, Slovenia) at 60 °C and 40 mm Hg until a constant weight was achieved. Soluble solids (SS, %) were determined at 20 °C by using of a refractometer (Carl Zeiss, Oberkochen, Germany). The pH was measured by using a pH meter (Mettler Toledo, Switzerland). The total acidity (TA, %) was measured by titration with 0.1 N NaOH, based on citric acid [27]. The total (TS, %) and the reducing sugars (RS) were determined by the Luff–Schoorl method [27]. Pectic compounds (PC, %) were determined by the Carré–Haynes method [28]. All measurements were carried out in duplicate.

Determination of ascorbic acid, total phenolic compounds, antioxidant activity and colour measurement are described later (Section 2.4.2, Section 2.4.3, Section 2.4.5 and Section 2.4.6, respectively).

### 2.3. Preparation of Rose Hip Nectar Formulations and HVED Treatment

In order to prepare the puree, rose hip pulp was diluted with water in equal proportions. The rose hip puree was obtained using an electric mill with a 1 mm pore size sieve. All nectar samples were prepared from freshly mashed rose hip puree and they contained a min 40% of puree (in accordance with current Croatian legislation) [29]. To provide nectar with 12.5% of soluble solids and a pH of 3.7, basic nectar formulations were prepared by the addition of fruit puree (40%), sucrose (7.5%), 10% solution of citric acid and water. The distilled water and the sugar used to prepare the nectars were microbiologically tested before use. The same basic nectar composition was used to study the effects of different HVED conditions (frequency and time) on physicochemical parameters, antioxidant activity and the colour parameters of rose hip nectar.

The integral parts of the HVED instrument, that was custom made by Ingeniare CPTS 1 (Osijek, Croatia) are: a high-voltage DC generator, an energy tank (capacitor), a high-voltage switch, a chamber and an automatic control unit to control the pulse frequency, time of treatment and speed of mixing. During the treatment, electrodes were attached to the electrode carrier with the ability to adjust the distance between the electrodes (20 mm). A needle with a diameter of 2.5 mm and a grounding electrode with a diameter of 45 mm were immersed in the nectar during the treatment.

After preparation and homogenisation, the nectar samples (300 mL) were treated with the HVED instrument. (The treatments were performed by high voltage electric discharge (30 kV) at two different frequencies (50 Hz,100 Hz) and three processing time intervals (10, 15, 20 min), with mixing on a magnetic stirrer, which is an integral part of the chamber. The control sample was not treated. All the treatments were done in duplicate.

Basic nectar formulations were prepared for samples N and NP. Unlike the sample NP, the sample N was not treated with HVED. The HVED treatments were conducted at 100 Hz for 20 min. The preparation of NBP 1 and NBP 2 samples included microwave blanching of pulp and puree, respectively. A portion of the pulp and the puree (in 100 g cycles) were blanched for 45 s at 700 W (KOR-63A5 Daewoo, Seoul, S. Korea). In addition, a low-calorie nectar (NSP) was also prepared. To prevent a reduction in the viscosity and the mouthfeel of low-calorie samples, hydrocolloid pectin was added. The solution of pectin was prepared by dissolving a certain mass fraction of pectin in distilled water, heating and then agitating with a magnetic stirrer. The nectar was processed by adding the puree (40%), a low-calorie sweetener (3.75%), 10% solution of citric acid, 0.2% pectin and water up to 8% soluble solids and pH 3.7. The pasteurised nectar (NPA) was prepared by pasteurisation at 85 °C for 20 min. The nectar was placed in a water bath maintained at 97 °C and the sample achieved 85 ± 2 °C after 8 min. All preparations were done in duplicate.

The samples were packaged in sterile 100 mL glass bottles immediately after processing. Physicochemical analyses were provided after preparations and microbiological analysis, during twelve days of refrigerated storage at 4 °C („0, 6th and the 12th day).

### 2.4. Analysis

Some analyses of the rose hip pulp were described in Section 2.2.

#### 2.4.1. Measuring of Electrical Conductivity (EC)

The electrical conductivity of nectars was measured at room temperature (23 °C) with an EC meter (Hanna Edge, Hanna Instruments, Woonsocket, RI, USA). The EC values are expressed in µS/cm. The measurements were done in duplicate.

#### 2.4.2. Ascorbic Acid Determination

Ascorbic acid (L-AA content) in rose hip pulp and nectars (AOAC method) was measured using 2,6-dichlorophenol indophenol titration. The results were expressed in mg/100 g. The measurements were performed in duplicate.

#### 2.4.3. Extraction and Determination Total Phenolic Compounds (TPC)

A sample (1 g) was extracted in a 10 mL mixture (water and ethanol, 80:20) at 20 °C for 15 min in an ultrasonic bath (DT 52 H, Bandelin, Berlin, Germany). After filtration, the extract was used for the determination of TPC and antioxidant activity.

The total phenolics were evaluated using the Folin–Ciocalteu method [30]. A total of 0.2 mL of extract and 1.8 mL of deionised water were mixed with 10 mL (1 in10 dilutions with water) of Folin–Ciocalteu reagent and 8 mL of 7.5% solution of sodium carbonate. The colour was developed after 120 min, and the absorbance was read at 765 nm by a spectrophotometer (Cary 60 UV-Vis, Agilent Technologies, Santa Clara, CA, USA). The measurements were carried out in triplicate and the values were interpolated on a gallic acid calibration curve. The results were expressed as g of gallic acid equivalents per 100 mL (GAE/100 mL).

#### 2.4.4. Extraction and Determination of Flavan-3-ols and Flavonols

Approximately 0.2 g of homogenised rose hip nectar was weighed in a tube and a 1.5 mL of a methanol and water mixture (80:20) was added. The mixture was vortexed and extracted using an ultrasonic bath (Bandelin Sonorex RK 100, Berlin, Germany) for 15 min. After 10 min of centrifugation at 10,000 rpm (Eppendorf Minispin, Hamburg, Germany), the extract was pipetted into a plastic tube. The residue was extracted again (0.5 mL, 80% methanol) using the same procedure. The two extracts were combined, filtered (0.22 µm PTFE syringe filter) and analysed.

The extracts were analysed on the high performance liquid chromatography (HPLC) system 1260 Infinity II (Agilent Technology, Santa Clara, CA, USA) consisting of a quaternary pump, a photodiode array detector (PDA), a vial sampler, a Poroshell 120 EC C-18 column (4.6 × 100 mm, 2.7 µm) and a Poroshell 120 EC-C18 4.6 mm guard column. The mobile phases were 0.1% H_3_PO_4_ (mobile phase A) and 100% methanol (mobile phase B). The gradient was: 0 min 5% B, 5 min 25% B, 14 min 34% B, 25 min 37% B, 30 min 40% B, 34 min 49% B, 35 min 50% B, 50 min 51% B, 52 min 80% B, 54 min 80% B, 56 min 5% B and 58 min 5% B. The flow was set to 0.5 mL/min. A volume of 10 µL of samples was injected into the system. Flavan-3-ols and flavonols were identified according to UV/VIS spectra from 190 to 600 nm, and they were expressed as total flavan-3-ols and total flavonols by using calibration curves of (+)-catechin and quercetin-3-galactoside, respectively. Duplicate measurements from the same sample were carried out.

#### 2.4.5. Antioxidant Activity

The ABTS (2,2′-azinobis(3-ethylbenzothiazoline sulfonic acid)) assay followed the method of Cano and Arnao [31], with some modifications. A total of 0.2 mL of extract was mixed with 3 mL of ABTS reagent (2,2′-azinobis [3-ethylbenzothiazoline-6-sulfonic acid]-diammonium salt) and left in the dark for 90 min. The absorbance was read at 734 nm. For blank, the sample was replaced with water.

The antioxidant activity of the samples was also determined by the DPPH method using 2,2-diphenyl-1-picrylhydrazyl radical as described by Brand-Williams et al. [32] but with a slight modification. The amount of 0.2 mL of extract was mixed with 2.8 mL of DPPH solution (0.5 mM). After 15 min, the absorbance was read at 517 nm. Antioxidant activity was calculated from the calibration curve with Trolox as the standard (µmol TE/100 mL). All measurements were carried out in triplicate.

#### 2.4.6. Colour Measurement and Total Colour Change

The colour measurement of the pulp and nectar samples (in a glass cuvette located in the accessory CR-A70) was performed by Minolta CR-300 Chroma Meter (Konica Minolta Holdings, Inc., Tokyo, Japan). The measurements were performed in five repetitions and expressed in the L* a* b* and L* C* °h systems. The data are expressed as lightness (L), redness (a) and yellowness (b), chroma (C), intensity of colour or colour saturation value and hue angle (°h), from 0° for red, over 90° for yellow, 180° for green, up to 270° for blue and back to 0°.

The total colour change (∆E) was calculated from the L*, a*, and b* parameters according to Equation (1), where L_0_, a_0_ and b_0_ represent the values of the control (untreated) sample or values before HVED treatment, and L, a, and b are the values thereafter [33].
(1)$$∆E=\sqrt{{\left(L-L_{0}\right)}^{2}+{\left(a-a_{0}\right)}^{2}+{\left(b-b_{0}\right)}^{2}}$$

### 2.5. Microbiological Analysis

All of the samples were analysed in laboratories accredited according to the HRN EN ISO/IEC 17025 standard. The samples were serially diluted and inoculated on suitable culture solid media, according to ISO standards determined for each microorganism [34,35,36,37,38,39,40]. Before taking the analysis, each sample was mixed with a stomacher to ensure a homogeneous distribution of microorganisms. The following methods were used: aerobic mesophilic bacteria (AMB) *Enterobacteriaceae* (EB), *Escherichia coli* (EC), *Salmonella* spp. (S), *Listeria monocytogenes* (LM) and yeasts and moulds.

The analyses of bacterial pathogens (*Listeria monocytogenes* and *Salmonella* spp.) were performed using a method for detection that required 25 mL of the sample. The analyses for aerobic mesophilic bacteria, *Enterobacteriaceae* and *Escherichia coli* were preformed using enumeration methods that required 10 mL of the sample. The pour plate technique was used in duplicate for each serial dilution and all enumeration methods.

The measured water activity in all nectars was 0.95. The measurements were in duplicate, using a thermohygrometer (HygroPalm HP23-AW, Rotronic AG, Bassersdorf, Switzerland) at 20 °C.

### 2.6. Statistical Analyses

The results were expressed as the mean values ± standard deviation. Data were analysed by analysis of variance (ANOVA) and the post-hoc Tukey test, with the significance defined at *p* < 0.05. All statistical analyses were carried out using the software program Minitab 19.1 (Minitab LLC., State College, PA, USA) and Microsoft Excel (Office Professional Plus 2019, Microsoft, Redmond, WA, USA).

## 3. Results and Discussion

### 3.1. Characteristics of Rose Hip Pulp

Table 1 presents the values for the composition, antioxidant activity and colour of the rose hip pulp. These findings are consistent with the literature [3,41,42]. The exception is the sugar content (TS and RS), which was lower than the literature values (approximately 15%, with sucrose content of about 2%).

Rose hip has a high content of ascorbic acid, which is an essential component of vitamin C. Literature data regarding rose hip from the section *Caninae* shows variations in ascorbic acid content. By comparison to other species such as *Rosa drumalis* and *Rosa villosa, Rosa canina* L. usually has a low ascorbic acid content (510 mg/100 g) [4]. According to Demir and Ozcan [43], rose hip can have a vitamin C content of up to 2712 mg/100 g, while according to studies by Kazankaya et al. [44] the content ranged from 301 to 1183 mg/100 g. There was considerable variability between vitamin C levels in the Republic of Croatia. The results gained by Stanić [42] showed that the amount of vitamin C in rose hip fruit from different cultivations ranged from 79.26 (fruit of wild genotypes) to 341.92 mg/100 g of fresh rose hip (fruit from conventional cultivation). Brkić [45] found that the content of ascorbic acid was 406.70 mg/100 g of fresh rose hip *Rosa canina* L. In this work, the ascorbic acid content in the pulp was 500.74 mg/100 g FW. The differences between the studies are due to the variety, diverse growing conditions, altitudes, environmental factors and times of rose hip harvest. The reduction in content may also be due to the level of oxygen in the environment, the amount of light, and changes in endogenous plant growth regulators and temperature [46].

Polyphenols are important ingredients with positive biological activity, of which antioxidant activity is the most pronounced. The concentration of TPC in the pulp was 1.24 g GAE/100 mL, which is in agreement with other results. According to the results of Stanić [42], the highest concentration of TPC was found in organically grown fruit (1310.79 mg GAE/100 g), a slightly lower concentration in fruit from wild genotypes (1210.19 mg GAE/100 g), and the lowest concentration was found in fruit from conventional cultivation (1109.62 mg GAE/100 g). Yoo et al. [47] recorded total phenols in the amount of 818 mg GAE/100 g of fruit.

The antioxidant activity of pulp determined by the ABTS method was 138.451 µmol TE/100 mL and 106.930 µmol TE/100 mL (DPPH method). High values of antioxidant activity were also demonstrated by other authors [48,49].

### 3.2. Effect of HVED Treatment (Frequency and Time) on Physicochemical Properties of Rose Hip Nectar

The results of physicochemical parameters, antioxidant activity and colour characteristics of untreated (control) and HVED treated rose hip nectar are presented in Table 2 and Appendix A.

The plasma-generated high voltage discharge treatment caused a small decrease in pH (from 3.7 to 3.6, or 3.5 in treatments performed at 100 Hz). Almeida et al. [16] showed decreasing pH values in prebiotic orange juice after treatment with plasma (1min at 50 Hz) from 4.4 to 3.9. Illera et al. [50] confirm a decrease in the pH of all plasma-treated apple juice samples. After 4 and 5 min of treatment, the pH decreased from 3.73 to 3.59. After 28 storages, the pH value remained the same as when measured immediately after plasma treatment. Thirumdas et al. [51] explained that this effect occurs during plasma treatment due to the formation of chemical compounds, such as hydrogen peroxide, which contribute to the acidity of liquids. The change in the acid content of the juices may be due to the melting of the hydroxyl radical formed during plasma treatment. In addition, changes in pH can result due to the formation of nitric and nitrous acids [15].

The HVED treatment caused an increase in electrical conductivity (EC) in all samples, due to the formation of numerous charged particles (positively and negatively charged ions). The highest conductivity values were recorded for treatments performed at 100 Hz. Since pH decreased and EC increased, they showed a high negative correlation (Appendix A). Grymonpré et al. [52] explained that higher values of electrical conductivity result in a decrease in the rate of formation of reactive chemical species, which ultimately results in a weaker impact on nectar quality.

Rose hip nectar is a rich source of vitamin C, i.e., ascorbic and dehydroascorbic acid, which makes it an excellent antioxidant. The HVED treatment did not cause ascorbic acid degradation, even though the longest treatment lasted for 20 min. In fact, the amount of ascorbic acid significantly increased after a 20 min treatment with 50 and 100 Hz. There was no significant increase in the nectar’s temperature during the treatment, which may be the reason for stability, even after processing. Moreover, treatment with plasma might have inactivated the antioxidative enzymes responsible for destruction of ascorbic acid [26]. Shi et al. [26] also did not record significant degradation of vitamin C in freshly squeezed orange juice after plasma treatment. However, in most studies with fruit juices, significant degradation of ascorbic acid occurred during plasma treatment [13,15,53].

Plasma treatment had a positive effect on stability, with TPC rising to 14%. After plasma treatment, Dasan and Boyaci [24] also recorded an increase in the concentration of TPC in cherry juice (10–15%). Herceg at al. [54] obtained a similar trend in pomegranate juice, in which TPC increased by 15–49% after treatment (increase in the concentrations of ellagic, chlorogenic and ferulic acids, as well as catechins and punicalagin).

Furthermore, the values of flavan-3-ols and flavonols also increased after HVED, and the largest increase occurred at 100 Hz treatment for 20 min. The increase was significant for flavan-3-ols after the treatment with 100 Hz and for flavonols after the treatment with 50 Hz. Similar findings were obtained in previous studies with pomegranate juice, chokeberry juice, cashew apple juice and white grape juice, where plasma treatment positively influenced the content of total flavonols, namely phenolic compounds [17,54,55,56]. The increase in concentration could be explained by the fact that plasma promotes mass transfer and the release of TPC. A large number of chemically reactive particles are generated during the plasma treatment, which are assumed to have sufficient energy to destroy covalent bonds and thus stimulate numerous chain reactions [54].

It can be seen from results in Table 2 that nectars after HVED treatments showed a significant reduction in AA activity when compared to the control (untreated) sample. The lowest activity with used methods (ABTS and DPPH) was recorded at a lower frequency (50 Hz) and at the shortest treatment time (10 min). This decrease in results was in accordance with previous works with prebiotic orange juice [16], white grape juice [17] and cashew apple juice [56]. Based on the results of the research, it can be concluded that the product, plasma source, method and processing conditions are key factors, considering the effects on the antioxidant activity of the product [11].

The strongest Pearson correlation between colour parameters L* and a* was observed (Appendix A). Plasma treatment caused a significant decrease in L* (darker) and a* (less red) colour parameters in all samples.

When comparing the physicochemical parameters, antioxidant activity and colour characteristics of non-treated rose hip nectars (control) and rose hip nectars treated with 50 and 100 Hz, some statistically significant differences can be seen (Appendix A). Electrical conductivity was higher after the treatment with 100 Hz, compared to 50 Hz and the control. The ascorbic acid content was also statistically higher after the treatment with 100 Hz, compared to 50 Hz and the control, but after 20 min. The number of total phenols increased after the treatment with both 50 and 100 Hz in comparison to the control but after 20 min. There are no significant differences in flavan-3-ol or flavonol content in nectars treated with 50 or 100 Hz, although the amount of flavonols increased after both 50 and 100 Hz treatment in comparison to the control (at 10 and 20 min). Antioxidant activity was significantly lower after the treatment with both 50 and 100 Hz in comparison to the control. Colour parameters were statistically lower after treatment with 50 and 100 Hz in comparison to the control, but there are no significant differences between the treatment with 50 and 100 Hz. In addition, the results were further analysed with ANOVA to see the influence of the frequency of HVED treatment and time, or their interactions, on some parameters (Appendix A). The increase of electrical conductivity and the amount of flavan-3-ols were significantly influenced by both frequency and time or their interactions. The time of treatment had a significant influence on the increase of ascorbic acid and total polyphenol content. Lower antioxidant activity measured with ABTS can be connected to the significant influence of frequency and time as well to their interactions. Even though colour parameters were lower, frequency and time did not show a significant effect.

### 3.3. Influence of HVED Treatments on Physicochemical Properties of Rose Hip Nectar Formulations

A quality-based comparison of the HVED treatment of rose hip nectar was done with thermal pasteurisation. An HVED treatment of rose hip nectar with different formulations was conducted at 100 Hz for 20 min. The results of physicochemical parameters and antioxidant activity of different nectar formulations are presented in Table 3.

It is well known that conductivity has a decisive role in plasma generation. At applied voltage, a higher EC results in a shortening of the discharge channels, higher amplitudes of the discharge current and a higher sparking voltage [57]. Degradation of phenol by reactive oxygen species decreased with the increased conductivity. Thus, NSP nectar with the highest conductivity value (2790.3 µS/cm) had the highest content of phenolic compounds (0.223 g GAE/100 mL).

The effect of cold plasma on the stability of vitamins was mostly studied in relation to vitamin C [15]. It is a bioactive compound sensitive to different processing conditions (heat, oxygen, UV); hence, it is generally used as an indicator of quality and a criterion for valuable components [6]. The results presented in Table 4 showed a loss of ascorbic acid in pasteurised nectar (NPA) in the amount of 54%, as well as in the nectar NBP 2 (20%) that was prepared from blanched puree.

The content of phenolic substances was slightly reduced in the thermally treated samples (NBP 1 and NBP 2), and the largest decrease in TPC content (40%) was recorded in pasteurised juice (0.128 g/100 mL). This is likely due to the thermal degradation of phenolic compounds. Heat treatment can cause the dissociation of conjugated forms of phenolic acids (gallic, ferulic and *p*-coumaric acid) into free forms, which is manifested by a decrease in content. A similar trend was shown by the values obtained for flavan-3-ols and flavonols.

Pasteurised nectar had the lowest AA in this case as well, while the highest was in untreated nectar. The overall decrease of AA, regardless of the method, can be explained by the ability of present antioxidant compounds to scavenge the free radicals generated with HVED, which would decrease their concentration in the nectar [24]. The AA was determined with both methods and it showed a high correlation (Appendix A). However, the effect of HVED treatment on antioxidant activity of nectars requires further research.

The results of the colour measurements of nectars and the calculation of the total colour change (ΔE) relative to the untreated sample are presented in Table 4. The minimal colour change was recorded for NBP 2 nectar. This proves the positive effect of blanching on colour preservation. According to Cserhalmi (33), results were expressed as a relationship between ΔE and the human perception of colour. The total colour change (ΔE) was “not noticeable” (0–0.5) for NBP 2, “noticeable” (1.5–3) for NBP 1, and “well visible” (3–6) for the NP, NSP and NPA samples. The Pearson correlation coefficient showed strong relationships between the colour parameters of nectars (Appendix A).

### 3.4. Influence of High Voltage Electrical Discharge (HVED) on Microbiological Quality of Rose Hip Nectar Formulations

Today, the research on the use of plasma is mainly aimed at the destruction of microorganisms on the surface of food and packaging materials but also increasingly at the inactivation of microorganisms in juices and nectars. One of the objectives of this work was to study the microbiological quality of nectars treated with HVED. For this purpose, the compliance of the prepared nectar samples with the recommended microbiological criteria was studied. Nectars were analysed according to the criteria in the European Commission Regulation (EC) No 2073/2005 on microbiological criteria for foodstuffs and the criteria of the Microbiological Guidebook for Food (Ministry of Agriculture, 3rd modified edition, 2011) [58], whose implementation is regulated by the Food Hygiene and Microbiological Criteria for Foodstuffs Act (NN No 81/2013).

The analyses included aerobic mesophilic bacteria (AMB), *Enterobacteriaceae* (EB), *Salmonella* spp. (S), *Escherichia coli* (EC), *Listeria monocytogenes* (LM) yeasts and moulds. Analyses were performed immediately after nectar preparation—zero day (“0”) and after 6 and 12 days of storage at 4 °C. Aerobic mesophilic bacteria and *Enterobacteriaceae* were below the limit of quantification by the culture methods (<10 cfu/mL) (Appendix A).

The results in Appendix A show that *Salmonella* spp. (S) and *Listeria monocytogenes* (LM) were not detected in any of the tested nectars during the 12-day storage period. All results for *Escherichia coli* were below the limit of quantification by the culture methods (<10 cfu/mL) (Appendix A). The intentional inoculation of rose hip nectars with pathogenic bacteria has not been studied. Compared to bacteria, yeasts and moulds are more resistant to the plasma treatment due to more rigid cell walls made of chitin [59]. The results of microbiological tests showed that the number of yeasts and moulds exceeded the recommended limit for unpasteurised fruit juices (10^2^ cfu/mL). This means that the results do not comply with the recommended criteria of the Microbiological Guidebook for Food. Pasteurised nectar (NPA) and NBP 2 nectar (prepared from blanched puree) showed the best microbiological stability (<10 cfu/mL) after 12 days, in terms of yeast and mould counts (Figure 1).

The main components of the yeast and mould cell membranes, along with polysaccharides, are lipids and proteins, which can be affected by plasma reactive species, leading to the formation of fatty acid peroxide and oxidation of proteins that result in cell death. The inactivation mechanism of HVED is still elusive and unexplained, but there are several possible pathways of microbial inactivation. Energetic electrons generated by electrical excitation transfer their energy to neutral gas atoms in processes that generate chemically charged reactive species, which can accumulate on the outer cell membrane and cause its rupture [60]. Plasma forms OH radicals that can react with nearby molecules, leading to chain oxidation and destruction of DNA [61].

In a study by Lacombe et al. [62], plasma treatment was also not successful in reducing the total number of yeasts and moulds on blueberries, as opposed to bacteria.

Plasma treatment causes various types of stress in the cells of microorganisms, including yeast. Although some effects of plasma are independent, such as direct chemical damage to several cellular components by reactive plasma particles, there is still insufficient data on the relationships and correlations between the various effects of plasma on microorganisms. Because HVED is a relatively new technology, further studies are needed to determine the efficacy of the treatment in decontaminating juices and inactivating a wider range of microorganisms in different products.

## 4. Conclusions

In this study, the effect of high voltage electrical discharge (HVED) on the physicochemical properties and microbiological quality and safety of rose hip nectar formulations was investigated. The novelty of this study addressed using HVED treatment of rose hip nectars, including low-calorie, where nectar formulations showed significant stability under the different process conditions. The physicochemical analysis proved that the rose hip pulp and the prepared nectars are valuable sources of polyphenols and ascorbic acid as bioactive compounds, with high antioxidant activity. In terms of quality parameters, the highest degree of degradation was observed in pasteurised nectar. After storage, all nectars were in accordance with current microbiological criteria, except for the presence of yeasts and moulds. In addition to the pasteurised sample, HVED-treated rose hip nectar prepared from microwave-blanched puree with extended shelf life had a satisfactory microbiological safety after 12 days of refrigerated storage.

## Figures and Tables

**Figure 1 foods-11-00651-f001:**
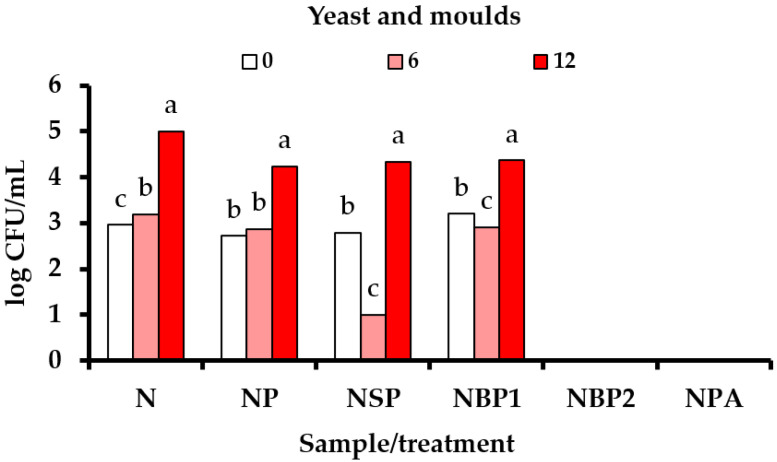
The average counts (log CFU/mL) of yeast and moulds in rose hip nectars during 12 days of refrigerated storage. Data were analysed by post-hoc Tukey test, with the significance defined at *p* < 0.05.

**Table 1 foods-11-00651-t001:** Characteristics of rose hip pulp.

Parameter	Content
total dry matter (TDM)	31.48 ± 0.35%
soluble solid (SS)	27.17 ± 0.40%
pH	3.50 ± 0.02
total acidity (expressed as citric acid) TA	3.14 ± 0.04%
ascorbic acid	500.74 ± 13.60 mg/100 g
total (TS) and reducing sugar (RS)	6.47 ± 0.27%; 6.25 ± 0.0%
pectin compounds (expressed as pectate) PC	11.79 ± 2.13%
total phenolic compounds (gallic acid equivalent)	1.24 ± 0.006 g GAE/100 mL
antioxidant activity (ABTS)	138.451 µmol TE/100 mL
antioxidant activity (DPPH)	106.930 µmol TE/100 mL
Colour parameters L*	30.02 ± 0.11
C*	34.17 ± 0.17
°h	46.50 ± 0.50

**Table 2 foods-11-00651-t002:** Physicochemical parameters, antioxidant activity, and colour parameters of untreated and HVED treated rose hip nectar.

F (Hz)	Sample/ Time (min)	pH	Electrical Conduct. (µS/cm)	Ascorbic Acid (mg/100 g)	TPC (g GAE/100 mL)	Flavan-3-ols (mg/kg)	Flavonols (mg/kg)	AA (µmol TE/100 mL)		Colour Parameter	
								ABTS	DPPH	L*	a*
0	control	3.70 ± 0	2223.0 ± 2.83 ^c^	97.57 ± 1.04 ^b^	0.212 ± 0.06 ^c^	487.76 ± 19.44 ^a^	22.96 ± 0.82 ^c^	28.62 ± 0.09 ^a^	15.85 ± 0.34 ^a^	37.43 ± 0.33 ^a^	10.93 ± 0.71 ^a^
50	10	3.60 ± 0	2256.0 ± 15.56 ^c^	104.11 ± 0.78 ^a^	0.234 ± 0.02 ^b^	513.16 ± 2.55 ^a^	31.01 ± 0.12 ^b^	22.89 ± 0.51 ^c^	12.70 ± 0.10 ^b^	34.02 ± 0.02 ^b^	8.16 ± 0.02 ^b^
50	15	3.60 ± 0	2427.5 ± 13.44 ^b^	100.41 ± 1.83 ^a,b^	0.187 ± 0.01 ^d^	494.03 ± 6.25 ^a^	33.69 ± 0.05 ^a^	25.29 ± 0.80 ^b^	13.35 ± 0.29 ^b^	33.73 ± 0.12 ^b^	8.00 ± 0.13 ^b^
50	20	3.60 ± 0	2506.5 ± 9.19 ^a^	106.51 ± 2.09 ^a^	0.278 ± 0.05 ^a^	521.72 ± 19.33 ^a^	30.31 ± 0.01 ^b^	25.35 ± 0.08 ^b^	13.28 ± 0.69 ^b^	34.13 ± 0.08 ^b^	8.13 ± 0.16 ^b^
0	control	3.70 ± 0	2223.0 ± 2.83 ^c^	97.57 ± 1.04 ^b^	0.212 ± 0.06 ^b^	487.76 ± 19.44 ^b^	22.96 ± 0.82 ^a^	28.62 ± 0.09 ^a^	15.85 ± 0.34 ^a^	37.43 ± 0.33 ^a^	10.93 ± 0.71 ^a^
100	10	3.50 ± 0	2513.0 ± 11.31 ^b^	105.03 ± 2.09 ^b^	0.237 ± 0.02 ^a^	481.71 ± 2.11 ^b^	32.19 ± 0.14 ^a^	26.08 ± 1.26 ^b^	15.17 ± 0.44 ^a,b^	34.19 ± 0.08 ^b^	8.21 ± 0.07 ^b^
100	15	3.50 ± 0	2549.5 ± 6.36 ^a,b^	99.48 ± 2.62 ^b^	0.217 ± 0.05 ^b^	539.09 ± 3.48 ^a^	29.62 ± 7.56 ^a^	26.97 ± 0.66 ^a,b^	13.49 ± 0.98 ^c^	34.21 ± 0.07 ^b^	8.29 ± 0.06 ^b^
100	20	3.50 ± 0	2585.0 ± 25.46 ^a^	113.72 ± 1.82 ^a^	0.236 ± 0.02 ^a^	550.90 ± 4.97 ^a^	32.23 ± 0.18 ^a^	25.07 ± 0.55 ^b^	14.09 ± 0.06 ^b,c^	34.19 ± 0.06 ^b^	8.25 ± 0.24 ^b^

Means ± standard deviation (SD) in the same column with different letters are significantly different (post-hoc Tukey test *p* ≤ 0.05). Flavan-3-ols—as a (+)-catechin. Flavonols—as a quercetin-3-galactoside. GAE—gallic acid equivalent. TE—Trolox equivalent.

**Table 3 foods-11-00651-t003:** Physicochemical parameters and antioxidant activity of different nectar formulations.

Sample	Electrical Conduct. (µS/cm)	Ascorbic Acid (mg/100 g)	TPC (g GAE/100 mL)	Flavan-3-ols (mg/kg)	Flavonols (mg/kg)	AA (µmol TE/100 mL)	
						ABTS	DPPH
N	2223.0 ± 2.00 ^f^	96.30 ± 0.75 ^b^	0.189 ± 0.04 ^b^	491.273 ± 24.52 ^a^	37.285 ± 0.02 ^a^	40.930 ± 1.14 ^a^	17.917 ± 0.74 ^a^
NP	2651.7 ± 9.50 ^b^	111.69 ± 3.14 ^a^	0.186 ± 0.01 ^b^	487.757 ± 19.44 ^a,b^	22.961 ± 0.82 ^d^	29.703 ± 1.48 ^c^	15.57 ± 0.32 ^b^
NSP	2790.3 ± 11.5 ^a^	96.71 ± 1.31 ^b^	0.223 ± 0.02 ^a^	371.607 ± 4.43 ^c^	29.187 ± 0.09 ^b^	26.049 ± 1.03 ^d^	14.863 ± 0.36 ^b^
NBP 1	2600.0 ± 2.00 ^c^	105.96 ± 1.31 ^a^	0.163 ± 0.01 ^c^	430.119 ± 3.39 ^b,c^	30.261 ± 0.20 ^b^	27.330 ± 0.84 ^c,d^	14.745 ± 0.55 ^b^
NBP 2	2339.7 ± 1.15 ^e^	77.15 ± 2.80 ^c^	0.169 ± 0.08 ^c^	479.875 ± 14.81 ^a,b^	26.245 ± 1.22 ^c^	33.066 ± 0.43 ^b^	17.273 ± 1.03 ^a^
NPA	2508.7 ± 0.58 ^d^	44.20 ± 1.83 ^d^	0.128 ± 0.03 ^d^	401.884 ± 13.57 ^c^	17.398 ± 0.00 ^e^	14.621 ± 0.89 ^e^	8.969 ± 0.25 ^c^

Means ± SD in the same column with different letters are significantly different (post-hoc Tukey test *p* ≤ 0.05). Flavan-3-ols—as a catechin. Flavonols—as a q-3-galactoside. GAE—gallic acid equivalent. TE—Trolox equivalent. N—nectar without HVED treatment (control). NP—HVED treated nectar. NSP—HVED treated low-calorie nectar. NBP 1—nectar prepared from blanched pulp + HVED. NBP 2—nectar prepared from blanched puree + HVED. NPA—pasteurised nectar.

**Table 4 foods-11-00651-t004:** Colour parameters (L*, a*, b*, C*, °h) and total colour change (ΔE) of different nectar formulations.

Sample	Colour Parameter					ΔE
	L*	a*	b*	C*	°h	
N	36.23 ± 0.06 ^a^	6.76 ± 0.16 ^b^	14.67 ± 0.05 ^a^	16.40 ± 0.61 ^a^	65.30 ± 0.70 ^a^	n.a.
NP	33.85 ± 0.13 ^c^	8.08 ± 0.05 ^a^	12.24 ± 0.21 ^c,d^	14.66 ± 0.19 ^b,c^	56.40 ± 0.30 ^d^	3.65
NSP	33.53 ± 0.01 ^d^	8.04± 0.06 ^a^	11.84 ± 0.06 ^d^	14.27 ± 0.16 ^c^	55.97 ± 0.06 ^d^	4.11
NBP 1	34.33 ± 0.03 ^b^	8.11 ± 0.15 ^a^	12.95 ± 0.29 ^b^	15.14 ± 0.10 ^b^	57.80 ± 0.10 ^c^	2.90
NBP 2	36.33 ± 0.11 ^a^	6.94 ± 0.23 ^b^	14.64 ± 0.17 ^a^	16.32 ± 0.13 ^a^	64.20 ± 0.40 ^a^	0.21
NPA	33.75 ± 0.07 ^c^	6.67 ± 0.07 ^b^	12.38 ± 0.16 ^b,c^	14.10 ± 0.13 ^c^	61.20 ± 0.36 ^b^	4.59

Means ± SD in the same column with different letters are significantly different (post-hoc Tukey test *p* ≤ 0.05). N—nectar without HVED treatment (control). NP—HVED treated nectar. NSP—HVED treated low-calorie nectar. NBP 1—nectar prepared from blanched pulp + HVED. NBP 2—nectar prepared from blanched puree + HVED. NPA—pasteurised nectar. n.a.—not applicable.

## Data Availability

The data presented in this study are available on request from the corresponding author.

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
