# Peer review of "The Effect of High Voltage Electrical Discharge on the Physicochemical Properties and the Microbiological Safety of Rose Hip Nectars"

_foods, 2022, doi:10.3390/foods11050651_

Round 1

Reviewer 1 Report

This research evaluated a high voltage electrical discharge (HVED) treatment as an alternative to preserving the quality and safety of rose hip nectars with satisfactory results when preparing nectars with microwave-blanched fruit puree. Methods seem appropriate and comprehensive. However, please clearly indicate the used experimental design for each section. In my opinion, a factorial design would be the best choice so results can also indicate if a factor, namely frequency or time, or their combination significantly affects any of the response variables. Otherwise, the effect of each factor is being studied separately.

Also, it would be enriching to note the difference of spoilage m.o. respect pathogenic m.o. in the discussion. Since this study lacks a control treatment of juice inoculated with pathogenic bacteria, the suitability of HVED against these m.o. deserves further attention. 

Particular comments.

Abstract. At a first glance, this sentence may be misleading “The main goal of this study was to combine a rapid and inexpensive non-thermal technique such as high voltage electrical discharge (HVED), and rose hip to preserve valuable bioactive compounds…” since it seems authors combined HVED and rose hip components to preserve the quality of something else. I would suggest rearranging this sentence for instance, “…was to propose a rapid and inexpensive non-thermal technique such as high voltage electrical discharge (HVED) to preserve valuable rose hip bioactive compounds…”

In my opinion, there are many unnecessary “etc.” through the introduction section.

Line 97-98. “typhimurium” is not a species, it should not be ictlized and it starts with a capital “T”.

Lines 163-170. Acronyms N, NP, and NBP are not defined before this paragraph. Please define them first. In this same paragraph is not clear to me why there is a different composition for the described nectar than that of lines 145-147.

Line 291. Check value because table 1 indicates 1.24 g/L while this line indicates 1.24 g/ 100 mL.

Reviewer 2 Report

It is advisable to include the statistical significance among different treatments for the microbiological assay.

Reviewer 3 Report

Authors describe the Effect of High Voltage Electrical Discharge on the Physico-chemical Properties and Microbiological Safety of Rose Hip 3
Nectars. The article is well organized and in my opinion is MAJOR REVSION. I reccomented that authors improve the experimental part by adding more antioxidant assay since a multi-target approach is now requested by the scientific community to test the antioxidant activity of sample. In fact they use just DPPH and ABTS test that are able to investigate the radical scavenging of sample. Move Tables 3, 6 and 7 in Supplementary. Insert Pearson correlation analysis between phytochemical content and antioxidant activity, and CIElab parameters. Since Hue was calculated a and b parameter should be removed by Table 1. improves discussion of antioxidant data after performing additional experiments. Rewrote conclusion section. Why this study is new?
